# Wish-Granting Interventions Promote Positive Emotions in Both the Short and Long Term in Children with Critical Illnesses and Their Families

**DOI:** 10.3390/children12010047

**Published:** 2024-12-30

**Authors:** Hannah Roberts, Jenny Cook, Apple Lee, Wei Kok Loh, Nigel Teo, Joanne Su Yin Yoong, Marguerite Gorter-Stam

**Affiliations:** 1Make-A-Wish Foundation International, 1217 WC Hilversum, The Netherlands; 2Research For Impact, Singapore 159964, Singapore; 3Close of Life, Graaf Zeppelinlaan 23, 1185 HC Amstelveen, The Netherlands

**Keywords:** pediatric palliative care, wish-granting interventions, well-being, quality of life, resilience

## Abstract

Background: Wish-granting interventions are recognized as positive experiences for children with critical illness and their families. While the positive effects have been shown in the immediate and short term, data on their long-term effects are lacking. Objectives: To evaluate the effects of wish-granting interventions on children and parents during and post intervention—both in the weeks after, and up to 5 years after—and to examine any differences between these two groups. Methods: A large-scale international survey was distributed to children (aged 13–17 years old in 2023) and their parents across 24 countries who received a wish-granting intervention in the preceding five years by Make-A-Wish Foundation International. Primary outcomes were positive emotions experienced by children and parents during and after the intervention (short term and long term). The secondary outcomes assessed were negative emotions in wish children, and to what extent children and parents felt the intervention created a sense of normalcy, benefitted other family members, created a happy memory, and gave relief from medical treatment, plus the perceived importance of wish-granting interventions. Results: The responses of 535 children and 1062 parents were analyzed. Both groups reported increases in positive emotions during the early intervention stages, peaking when the wish was granted and persisting in the short and long term. No significant differences were found between children and parents during wish-granting or after the intervention. Negative emotions were reported by a minority of children. Over 80% of children and parents felt the intervention created a happy memory and provided relief from their medical treatments. Nearly all children (96.8%) and parents (95.4%) viewed a wish-granting intervention as important for children with a critical illness. Conclusions: Wish-granting interventions can provide positive emotional benefits to both children and their families in both the short and long term.

## 1. Introduction

Critical illnesses—progressive, degenerative, or malignant medical conditions that place a child’s life in jeopardy—impact more than 13.7 million children and youth aged between 3 and 17 years around the world [1]. The impact of a life-changing condition on children and their families is commonly known and is profound as they face treatments associated with physical, mental, and social challenges, reducing the overall quality of life as efforts are made to cure or care.

Research has shown that wish-granting interventions for children who are experiencing life-threatening medical conditions can reduce anxiety and depression and increase hope and positive emotions [2,3]. In the context of wish-granting, these positive emotions can lead to improvements in mental and physical well-being. For instance, research has found that children who were granted wishes were more likely to experience a decrease in the number of unplanned hospital and emergency department visits [4]. In addition, improved social well-being, as indicated by increased scores on scales of gratitude, love, communication, and benefit-finding, were reported by both children receiving wishes and their parents [5]. More generally, this is also supported by wider research that suggests that psychosocial interventions for children with cancer can be effective in reducing anxiety and depression and improving the quality of life, having a positive impact on physical symptoms and well-being [6].

Emerging research has focused particularly on the effects of a wish intervention on parents and siblings, demonstrating a ‘ripple effect’ whereby these individuals also experience improvements in their mental health and well-being concurrently to the wish child [5,7,8]. Siblings can share benefits such as positive memories, greater sharing with the wish child, and greater assurance about their sibling’s illness [3]. Overall, wish-granting interventions can bring a range of benefits to the health and well-being of critically ill children and their families [9].

Despite the positive benefits, the literature also indicates a risk of unintended negative outcomes. Children with critical illness experience a broad range of emotions, including loneliness, isolation, and anxiety [10]. Parents and siblings can also be negatively affected as the wish experience may be a reminder of the negative feelings experienced regarding their child’s condition. The wish can serve as confirmation of the life-threatening nature of their child’s illness [7]. It may also highlight challenging family dynamics such as sibling jealousy or conflicts between the parents and the wish child on the choice of their wish [3].

Critical questions remain in terms of how the effects of a wish-granting intervention evolve over time and whether children and their parents experience these benefits differently. Much of the existing literature focuses on positive emotions and other psychological resources up to a period of weeks after the intervention, but potential lasting effects are less well documented [11].

To this end, the aims of this study were to evaluate the short- and long-term effects of a wish-granting intervention, and to explore any differences in the impact of a wish-granting intervention between the wish child and the parent/guardian.

## 2. Methods

### Study Design

Data on the impact of wish-granting were collected by means of a global survey of wish children and their parents or guardians, with the help of Make-A-Wish Foundation International.

Make-A-Wish Foundation International is the world’s largest wish-granting charity, having granted more than 585,000 wishes to date. The charity supports and oversees national offices (herein referred to as Affiliates) in 40 countries worldwide. When granting a wish, the charity follows the “Wish Journey”—a series of stages the child embarks upon from the point of application, namely Wish Capture, Wish Design, Wish Anticipation, Wish Realization, and Wish Effect. Briefly, this refers to capture of the desired wish from the child, the wish being developed and tailored for each child, small actions carried out to build excitement for the wish, the wish being granted, and the period after the wish has been granted. In this way, the charity aims to maximize the impact of a wish for both wish children and their families, both during and after the journey. All Affiliates adhere to the Wish Journey framework.

Make-A-Wish Foundation International Affiliates disseminated the survey to wish children aged 13–17 at the time of distribution who had received their wish in the five years prior to the date of dissemination, as well as to wish parents or guardians. The total number of individuals who received the survey could not be determined due to a lack of monitoring by Affiliates. The number of wishes granted in the last 5 years per Affiliate is provided in the Appendix A.

The survey was iteratively developed and reviewed by the Medical Advisory Committee of Make-A-Wish Foundation International, comprising 8 subject matter experts around the world, plus senior executives at 26 Affiliates who had expressed an interest in participating in the research. The surveys were made available in English, French, German, Greek, Italian, Portuguese, Portuguese (Brazil), Spanish, Spanish (Latin America), Arabic, Chinese (Simplified), Chinese (Traditional), Japanese, Korean, and Malay. The translations were completed by a mix of professional translators and Affiliate staff. A second native speaker, a staff member at the local Affiliate, subsequently reviewed the translation for clarity. A visual aid depicting the stages of the Wish Journey was included to facilitate comprehension.

The surveys were deployed online via a secure third-party international survey platform, SurveyMonkey. Separate survey links were set up for EU/UK and non-EU/UK respondents in light of General Data Protection Regulation (GDPR) requirements. The survey links were disseminated to all Make-A-Wish Affiliates via the Make-A-Wish Foundation International headquarters. Each Affiliate was responsible for sharing the links with eligible wish children and parents in their respective countries/territories and requesting that they complete the survey online. Due to a lack of literacy and/or internet access, four offices (Brazil, Colombia, India, and Shanghai) administered the survey to children and families in person or via telephone, and then entered the results into SurveyMonkey. Wish children were requested to complete the survey themselves, and to obtain their parent/guardian’s consent before responding. The survey links accepted responses for five months (from 5 May to 11 September 2023), at which point responses plateaued, and no further significant participation was expected.

Ethical approval for the conduct of this study was obtained from Singapore Management University (approval number IRB-23-054-A043(423)).

## 3. Outcomes

### 3.1. Primary Outcomes

The primary outcomes consisted of seven key emotions: joy and happiness, respite and distraction, self-efficacy, looking forward (broadening horizons), family bonding, inclusion and social engagement, and individual well-being. The survey design was informed by a comprehensive review of the literature on wish-granting interventions [12]. In particular, the primary outcomes were adapted from Heath et al. [13], which provides a framework for exploring the effect of wish-granting interventions on well-being. A description of each outcome was provided for respondents alongside the survey question (see Table 1).

All respondents were asked to what extent they experienced these feelings at each stage of the Wish Journey on a 5-point Likert scale, ranging from 1 (Not at all) to 5 (Very much). For sustained effects post-intervention, respondents were first asked how they felt if they “think back to the weeks after your wish came true” (short-term effect) and then how they feel “when you think about your wish today” (i.e., the day on which the survey was taken) (long-term effect).

### 3.2. Secondary Outcomes

Wish children were also asked to what extent they experienced negative emotions (sadness, disappointment, and frustration) at any point on the Wish Journey, using the same 5-point Likert scale.

In addition, respondents were asked about the perceived impact of the wish on their family. Both children and parents indicated their level of agreement with the following statements: “the wish created a lasting memory of a happy occasion”, “the wish created a sense of normalcy”, “the wish experience benefitted other members of the family” and “the wish gave the family relief from the my (child’s) medical care/treatment”. Lastly, respondents were asked to what extent they agreed with the statement “I think a wish experience is important for every child with a critical illness to have”. All questions were asked using a 5-point, ranging from 1 (Strongly disagree) to 5 (Strongly agree).

## 4. Statistical Analysis

Missing data were assessed for each variable prior to analysis. Cases with missing data for primary outcomes, date of birth, or age at which the wish was granted were excluded, as this prevented an understanding of the time since the wish was realized. Cases where the time since wish realization exceeded five years were removed due to predetermined exclusion criteria. Cases where the age of the child (*n* = 49) or the time since the wish was granted (*n* = 87) was implausible were also removed. Cases with missing data on other variables (i.e., other demographic variables and secondary outcomes) were retained in the analysis to maximize sample size.

Demographic characteristics are presented as the frequency (*n*) and proportion (%) of respondents within each demographic category. For primary outcome variables, the median and IQR were computed, as Shapiro–Wilk tests indicated that all primary outcomes were not normally distributed.

Statistical differences between children and parents in the primary outcomes were assessed using the non-parametric Wilcoxon rank-sum test. To account for multiple comparisons, Bonferroni correction was applied to adjust the *p*-values. Statistical significance was set at a corrected *p*-value of less than 0.05 for all tests. All statistical analyses were performed using R software (version 4.3.1).

## 5. Results

In total, 932 responses were received from wish children and 1858 responses from wish parents/guardians, representing 24 Make-A-Wish Affiliates (60% of the 40 Affiliates). The number of responses per country and region is available in the Appendix A. After excluding incomplete responses, the responses of 535 wish children and 1062 parents were analyzed.

### 5.1. Demographic Characteristics

Most wish children respondents were 17 at the time of the survey (37.2%) (Table 2), and more than half of wish children respondents had their wish granted in the last year.

For most parents of wish children (56.1%), it had been between 2 and 5 years since the wish of their child was granted (Table 3). The vast majority were a parent of the child, and mostly mothers completed the survey (73.16%).

### 5.2. Primary Outcomes in Wish Children

Figure 1 details the proportion of children experiencing the primary outcomes to some extent per Wish Journey stage. Table 4 details the number and proportion of children answering each response category per Wish Journey stage. A clear pattern is seen whereby the extent to which one of the outcomes is experienced climbs in the first and second stages of the Wish Journey. While the average proportion of children that report they “not at all” experience these emotions in the first stage is 6.5%, this drops to 1.3% in the Wish Anticipation stage. The Wish Realization stage stands out as a time during which these outcomes are experienced to the maximum. Almost 90% of children report feeling “joy and happiness” “very much” at this stage. All other outcomes are experienced for more than 70% of children “very much” as well, except for “inclusion and social engagement” which was felt by 64.4%. Small declines are seen in the short term, in the weeks after the wish took place. When thinking back to their wish from the day the survey was taken, “inclusion and engagement” was felt by 55% of respondents; all other emotions were experienced by more than 60% of respondents. “Joy and happiness” was felt “very much” by 80.1% of respondents.

### 5.3. Primary Outcomes in Wish Parents/Guardians

Figure 2 details the proportion of parents experiencing the primary outcomes to some extent per Wish Journey stage, and Table 5 shows the number and proportion of parents answering each response category per Wish Journey stage. A higher proportion of parents report experiencing all primary outcomes “very much” in the first stage (‘Wish Capture/Design’) compared to the wish children respondents (51.2% on average for parents compared to 40.3% of wish children). The average proportion that did not experience these emotions “at all” in the first stage is 2.7%, compared to 6.5% for wish children. The peak of these emotions is seen in the Wish Realization stage. Similarly to the wish children, all outcomes were experienced for more than 70% of parents “very much” as well, except for “inclusion and social engagement”. Looking back at the weeks after the wish was granted, around two-thirds of parents report feeling 6 of the 7 primary outcomes. “Inclusion and engagement” was felt by 57.4% of respondents. This drops very slightly looking back at the wish at the time of taking the survey. For “joy and happiness”, there is a small increase from 80% feeling this “very much” in the weeks after the wish to 81% feeling this when looking back at the time of the survey.

### 5.4. Secondary Outcomes

#### 5.4.1. Negative Outcomes in Children

Children were asked about potential negative experiences. A minority reported experiencing sadness (7.9%), disappointment (6.9%), and frustration (8.0%) “very much” at some point during their Wish Journey (Table 6).

#### 5.4.2. Impact on Wish Family

Over 95% of wish children and parents agreed that the wish experience created a lasting memory of a happy occasion (Table 7). Additionally, 43.7% of wish children and 51.0% of parents strongly agreed that the wish also created a sense of normalcy. Approximately half of both groups strongly agreed that the wish positively impacted other family members and provided their family relief from the child’s medical care or treatment. Among wish children, 74.0% strongly agreed that a wish experience is important for every child with a critical illness. This sentiment is stronger among parents, with 81.9% strongly agreeing on its importance.

#### 5.4.3. Differences Between Wish Children and Parents/Guardians

Wilcoxon tests were conducted to compare the primary outcomes per Wish Journey stage between wish children respondents and parent respondents. All tests in the Wish Capture/Design and Wish Anticipation stage are significant. Although the median scores are often the same (i.e., median = 4 or 5), the IQR is narrower for parents in many cases. Parents are therefore more consistent in their responses in these two stages.

All tests conducted per variable in each future stage, barring two, are non-significant. The medians and IQRs are uniform across both children and parents. These findings indicate that wish children and parents did not experience significantly different primary outcomes during the Wish Realization phase, or when reflecting on the wish in the short or long term (Table 8).

## 6. Discussion

### 6.1. Main Findings

Our findings show that positive emotions increased throughout the wish-granting intervention and peaked during Wish Realization, where over 70% of children and parents experienced most primary outcomes “very much”. In the short term after the wish, positive emotions slightly declined but remained high, with outcomes felt “very much” by an average 70.6% of children and 68.5% of parents. The results are consistent with the existing literature that shows that wish-granting interventions can provide positive, joyful experiences for children with life-threatening diseases and their families [7,11]. Looking back from the time of the survey (up to 5 years post-wish), in the long term, emotions remained strong, particularly “joy and happiness”, which was experienced “very much” by 80.1% of children and 81.0% of parents. It is known that pleasant events that are frequently discussed with other individuals demonstrate a low affective fading [14]. In addition, the affect associated with unpleasant events fades faster than the affect associated with pleasant events, a phenomenon referred to as the fading affect bias (FAB) [15]. This supports our findings that positive emotions remain high, not only in the short term, but also in the long term.

Parents had significantly higher median scores in the initial stages prior to Wish Realization; however, wish children and parents did not experience significantly different primary outcomes during Wish Realization or post wish. The benefits of a wish intervention on parents have been identified in previous research, wherein parents who participated in wish interventions reported improvements in their mental health and emotional well-being [7,9]. The differences between children and parents may be attributed to the long-term perspectives of parents compared to children who are more likely to live in the present. Further, a child may view the wish more narrowly as a personal event, whereas parents may frame the wish more widely in the context of their whole family. This is supported by our results that consistently show that parents more often “strongly agree” with the family-level statements. The broader framing may lead to parents experiencing positive emotions earlier in the Wish Journey.

In terms of the family-level outcomes, the findings in this study show that the wish intervention was perceived as a lasting memory of a happy occasion, able to create a sense of normalcy, and benefitted other family members and gave the family relief from the child’s medical treatment. This confirms prior research into caregiver perceptions of a wish intervention [3]. We know that children and parents report improvements in quality of life [2,3,6,7,9,10,15], improved resilience, and coping, including an increased ability to cope with their illness and an increased confidence and self-esteem [16]. Therefore, it was no surprise that wish children and parents in our study strongly agreed that a wish experience is important for every child with a critical illness to have.

Despite the positive benefits, negative emotions were reported by a minority of wish children. This has been highlighted by previous research indicating that an intervention may lead to conflicted emotions, as it can serve as a reminder of the child’s health status [7,10]. Other negative factors included the timing of the wish, barriers in accessibility (e.g., for wheelchairs), difficulties with travel, and family dynamics [3]. Some factors may be foreseen by careful planning (accessibility); however, not all can be eliminated. We believe that awareness of such factors can facilitate the removal of these barriers for future wish children and families.

Our results indicate that the wish intervention was almost universally regarded as important for children facing a critical illness. We know from the literature that significant improvements in physical health are found in wish children, including reduced nausea [9], increased energy levels post-wish [7], improved physical well-being [3,11,16], and improved functional skills [3,9]. Combining these results with our long-term effects, we can state that a wish-granting experience may translate into improvements in mental and physical well-being for both children and their families in the short and long term. This will aid in reducing the number of unplanned hospital and emergency department visits [3,4]. We therefore want to strongly appeal to clinicians to incorporate wish-granting experiences into their clinical routine.

### 6.2. Strengths and Limitations

Our study has several strengths. To our knowledge, this is the first study to examine the impact of wish-granting interventions across diverse international contexts, in contrast to prior research that has focused primarily on single countries and Western cultural settings. Additionally, we were able to quantitatively assess the experience of the wish intervention at multiple timepoints, both during the intervention and twice post-intervention, in wish children and parents. This approach enabled an understanding of the intervention’s sustained effects and provided insights into the differences in experience for children and their parents. Furthermore, this study benefits from a relatively large sample size compared to similar studies, which strengthens the robustness of the findings.

However, there are also several limitations to this study. While the survey was disseminated to 40 Affiliates, no responses were received from 12 of the 40, with response rates varying substantially across Affiliates, leading to a concentration of data from certain countries, therefore limiting generalizability. While the translation of the survey into multiple languages is a strength, the potential for translation inaccuracies cannot be ruled out. It is noted that the English version used a scale of ‘Not at all’, ‘Somewhat’, ‘Neutral’, ‘A little’, ‘Very much’, although there was no strong indication that this impacted the findings. Many respondents tended to answer consistently in the same direction, suggesting that the responses were clear and not muddled by the scale format.

The method of survey deployment also varied across Affiliates due to differences in internet access and literacy levels, with some using offline formats. In this way, we aimed to expand participation to typically underrepresented groups; however, these different methods may have introduced respondent bias. Moreover, the survey was deployed to children aged 13–17, while Make-A-Wish serves children aged 3–17. This was intended to address issues of consent and recall accuracy, but at the same time meant that certain perspectives were potentially underrepresented. The developmental characteristics of the children were also not taken into account in this study. Puberty is associated with changes in drives, motivations, psychology, and social life, and may influence the perception of a wish granted. Since puberty also varies in age, gender, and personality, this would be an interesting aspect to take into account when researching this age range. The generalizability of the findings may have also been affected by the exclusion of some cases based on missing data. Only cases with complete data on the primary outcomes were included, and excluded cases may differ systematically from those included.

Lastly, the post-intervention questions were framed generally, asking participants to reflect on their experiences in the “weeks after” the intervention, and looking back at the intervention from the date the survey was taken. The broad interpretation of this, alongside the variability in how long ago the Wish Realization was, may have affected the consistency of responses. As the survey was retrospective, recall bias may have influenced the accuracy of responses, potentially leading to over- or underestimation of their emotions during the Wish Journey. To eliminate certain forms of bias related to the retrospective nature of this study, we suggest a prospective study on this topic.

Furthermore, there are multiple factors that may influence the impact of a wish-granting intervention that have not yet been fully studied. These factors can be divided into patient-related factors, wish-related factors, and factors related to the environment of the child and family.

Examples of patient-related factors are the age, type of condition, and severity of the condition. For example, younger children may derive comfort and emotional support differently than adolescents, who might prioritize feeling a sense of normalcy. Similarly, the type of condition and the severity may influence how children and their families experience the intervention. Data on the illness of the child were not collected for European respondents in this study due to data privacy regulations, and this therefore highlights an area for potential future research. Additionally, the type of wish granted may play a role in the impact of the wish. A wish ‘to go’ somewhere may fulfill different needs in a child compared with ‘to have’ wishes (where the child typically receives material items). We speculate that a wish ‘to go’ may resonate more with an adolescent who wants more independence, whereas a younger child may respond more positively to tangible items that provide immediate joy.

Environmental factors include but are not restricted to the family’s circumstances, such as wealth and educational background. Families with greater financial resources or higher levels of education may perceive and experience the wish differently compared to families with fewer resources or limited educational opportunities. These factors could influence how families engage with and value the wish-granting process, as well as the extent to which it alleviates their emotional or practical challenges.

In addition to family circumstances, broader societal and regional factors also play a significant role. Cultural values, social norms, and healthcare practices differ widely across regions, shaping the way wish children and their families experience a wish. For instance, in some cultures, the collective family experience may be highly prioritized, while in others, the individual joy of the child might take precedence. Regional differences in healthcare infrastructure could further affect the perceived significance of a wish, as families in areas with limited medical resources might see the wish as a rare source of relief and hope, compared to families in regions with more comprehensive healthcare support. Similarly, the social safety net and the perceived perception of illness by their surroundings could be of great influence. Altogether, further research is warranted to explore the many factors that could influence the impact of a wish—including family circumstances (e.g., wealth and education), the child’s age, condition, and wish type, as well as societal and regional differences—in order to further understand and optimize the impact of wish-granting interventions.

## 7. Conclusions

This study finds that wish-granting interventions lead to positive emotions in the immediate, short, and long term that can translate into improvements in mental and physical well-being for both children and their families. This will aid clinicians to cure and care. This is underlined by our finding that most wish children and parents strongly agree that a wish experience is important for every child with a critical illness to have.

## Figures and Tables

**Figure 1 children-12-00047-f001:**
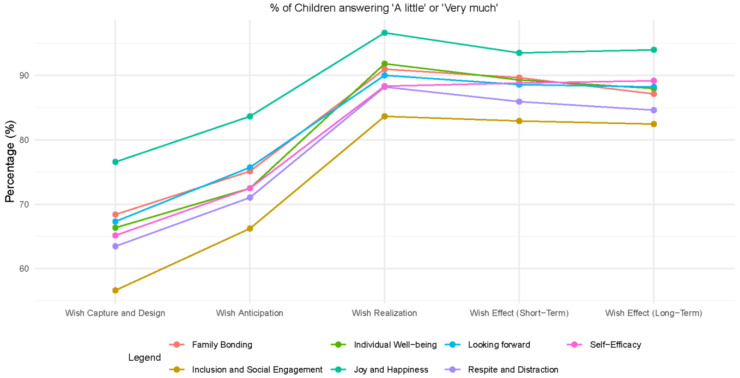
Proportion of children reporting experiencing each primary outcome “A little” or “Very much” per Wish Journey Stage.

**Figure 2 children-12-00047-f002:**
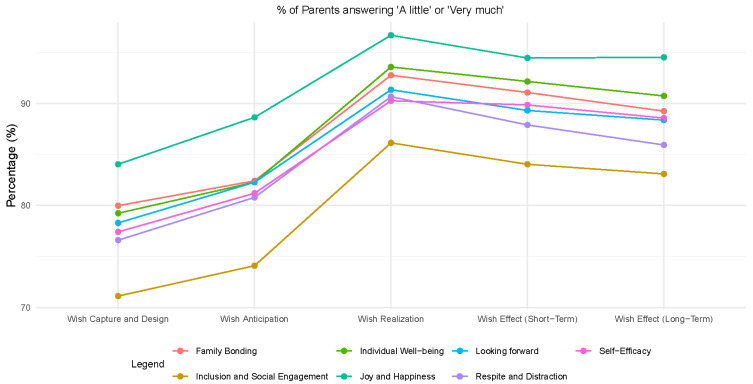
Proportion of parents reporting experiencing each primary outcome “a little” or “very much” per Wish Journey stage.

**Table 1 children-12-00047-t001:** Key outcomes identified in the literature.

Outcome	Theoretical Pathway	Description
Joy and happiness	Granting wishes fosters joy and happiness for children, their parents, and siblings by creating excitement and anticipation leading up to the wish, delivering the experience, helping children form positive memories, and providing a setting where the child feels uniquely valued.	Feel happy and excited
Respite and distraction	Wishes offer a welcome break from the child’s illness and treatment, inspiring them to engage with their treatment, ultiamtely supporting their health. For parents, wishes provide relief and distraction from concerns about their child’s condition	Feel like you could better cope with your condition
Self-efficacy	Wishes help foster a sense of growth and learning for the child.	Feel that you gained new knowledge, skills, and/or confidence
Looking forward (broadening horizons)	Wishes empower children and families to broaden their perspectives on what they can accomplish after the wish, encouraging them to live more fulfilling lives.	Feel that good things could happen in the future
Family bonding	Wishes promote a sense of family unity and normalcy, allowing families to spend meaningful time together during the wish experience and enhancing their overall coping abilities.	Feel closer to immediate family (e.g., parents, siblings, guardians)
Inclusion and social engagement	Wishes support children and families in forming deeper connections with their broader social networks and communities beyond the realms of health, education, and social care.	Feel closer to friends and/or community (e.g., relatives, neighbors)
Individual well-being	Wishes enhance the child’s overall resilience, boost confidence, and provide a sense of social and emotional support.	Feel socially and emotionally supported, be more resilient

**Table 2 children-12-00047-t002:** Demographic characteristics, child respondents.

	*n*	%
**Age**		
13	100	18.69
14	83	15.51
15	78	14.58
16	75	14.02
17	199	37.20
**Years since Wish Realization**		
0–1	298	55.70
2–<5	237	44.30
**Gender**		
Male	281	52.52
Female	245	45.79
Prefer not to say	1	0.19
Missing	8	1.5

**Table 3 children-12-00047-t003:** Demographic characteristics, parent respondents.

	*n*	%
**Gender**		
Both Mother and Father completed together	1	0.09
Female	777	73.16
Male	277	26.08
Prefer not to say	3	0.28
Missing	4	0.38
**Relationship to Child**		
Missing	9	0.85
Guardian/Ward	75	7.06
Parent	978	92.09
**Years since Wish Realization**		
0–1	466	43.88
2–<5	596	56.12

**Table 4 children-12-00047-t004:** Number and proportion of children reporting experiencing the primary outcomes, per Wish Journey stage.

	Do Not Remember	Not at All	Somewhat	Neutral	A Little	Very Much
	*n* (%)	*n* (%)	*n* (%)	*n* (%)	*n* (%)	*n* (%)
**Phase 1 and 2: Wish Capture/Design**						
Joy and Happiness	7 (1%)	47 (6.4%)	61 (8.3%)	65 (8.9%)	148 (20.2%)	405 (55.3%)
Individual Well-being	10 (1.4%)	48 (6.6%)	70 (9.6%)	116 (15.8%)	185 (25.3%)	303 (41.4%)
Family Bonding	7 (1%)	33 (4.5%)	75 (10.2%)	119 (16.2%)	183 (25%)	316 (43.1%)
Respite and Distraction	14 (1.9%)	48 (6.6%)	95 (13%)	113 (15.4%)	200 (27.3%)	262 (35.8%)
Inclusion and Social Engagement	7 (1%)	54 (7.4%)	99 (13.5%)	167 (22.8%)	181 (24.8%)	223 (30.5%)
Looking forward	7 (1%)	54 (7.4%)	69 (9.4%)	111 (15.1%)	187 (25.5%)	305 (41.6%)
Self-efficacy	6 (0.8%)	50 (6.8%)	82 (11.2%)	117 (16%)	223 (30.5%)	252 (34.5%)
**Phase 3: Wish Anticipation**						
Joy and Happiness	5 (0.7%)	5 (0.7%)	31 (4.2%)	87 (11.9%)	176 (24%)	428 (58.5%)
Individual Well-being	8 (1.1%)	10 (1.4%)	52 (7.1%)	133 (18.2%)	220 (30.1%)	308 (42.1%)
Family Bonding	9 (1.2%)	6 (0.8%)	43 (5.9%)	127 (17.3%)	229 (31.3%)	318 (43.4%)
Respite and Distraction	10 (1.4%)	10 (1.4%)	51 (7%)	146 (19.9%)	244 (33.3%)	272 (37.1%)
Inclusion and Social Engagement	13 (1.8%)	19 (2.6%)	60 (8.2%)	161 (22%)	230 (31.4%)	250 (34.1%)
Looking forward	7 (1%)	7 (1%)	47 (6.4%)	119 (16.2%)	229 (31.2%)	324 (44.2%)
Self-efficacy	11 (1.5%)	11 (1.5%)	47 (6.4%)	134 (18.3%)	246 (33.7%)	282 (38.6%)
**Phase 4: Wish Realization**						
Joy and Happiness	3 (0.4%)	1 (0.1%)	3 (0.4%)	19 (2.6%)	54 (7.4%)	653 (89.1%)
Individual Well-being	5 (0.7%)	5 (0.7%)	7 (1%)	41 (5.6%)	126 (17.2%)	548 (74.9%)
Family Bonding	6 (0.8%)	6 (0.8%)	10 (1.4%)	44 (6%)	96 (13.1%)	572 (77.9%)
Respite and Distraction	8 (1.1%)	12 (1.6%)	10 (1.4%)	53 (7.3%)	127 (17.4%)	520 (71.2%)
Inclusion and Social Engagement	5 (0.7%)	13 (1.8%)	23 (3.1%)	79 (10.8%)	140 (19.2%)	471 (64.4%)
Looking forward	6 (0.8%)	5 (0.7%)	5 (0.7%)	55 (7.5%)	98 (13.4%)	561 (76.8%)
Self-efficacy	8 (1.1%)	5 (0.7%)	13 (1.8%)	50 (6.9%)	124 (17.1%)	525 (72.4%)
**Phase 5: Short-Term Impact**						
Joy and Happiness	5 (0.7%)	3 (0.4%)	4 (0.5%)	28 (3.8%)	104 (14.2%)	588 (80.3%)
Individual Well-being	7 (1%)	5 (0.7%)	13 (1.8%)	51 (7%)	147 (20.1%)	509 (69.5%)
Family Bonding	8 (1.1%)	7 (1%)	8 (1.1%)	47 (6.4%)	124 (17%)	537 (73.5%)
Respite and Distraction	13 (1.8%)	7 (1%)	14 (1.9%)	64 (8.8%)	147 (20.2%)	484 (66.4%)
Inclusion and Social Engagement	11 (1.5%)	14 (1.9%)	16 (2.2%)	78 (10.7%)	159 (21.8%)	452 (61.9%)
Looking forward	8 (1.1%)	4 (0.5%)	8 (1.1%)	61 (8.3%)	122 (16.7%)	528 (72.2%)
Self-efficacy	12 (1.6%)	2 (0.3%)	8 (1.1%)	55 (7.5%)	136 (18.6%)	519 (70.9%)
**Phase 5: Long-Term Impact**						
Joy and Happiness	5 (0.7%)	2 (0.3%)	7 (1%)	27 (3.7%)	105 (14.3%)	586 (80.1%)
Individual Well-being	6 (0.8%)	6 (0.8%)	12 (1.6%)	56 (7.7%)	200 (27.5%)	448 (61.5%)
Family Bonding	9 (1.2%)	9 (1.2%)	10 (1.4%)	60 (8.2%)	174 (23.8%)	469 (64.2%)
Respite and Distraction	10 (1.4%)	9 (1.2%)	14 (1.9%)	76 (10.4%)	168 (23.1%)	451 (62%)
Inclusion and Social Engagement	8 (1.1%)	12 (1.6%)	18 (2.5%)	86 (11.8%)	205 (28%)	402 (55%)
Looking forward	6 (0.8%)	3 (0.4%)	11 (1.5%)	65 (8.9%)	160 (21.9%)	487 (66.5%)
Self-efficacy	8 (1.1%)	4 (0.5%)	9 (1.2%)	54 (7.4%)	176 (24.1%)	480 (65.7%)

**Table 5 children-12-00047-t005:** Number and proportion of parents reporting experiencing primary outcomes, per Wish Journey stage.

	Do Not Remember	Not at All	Somewhat	Neutral	A Little	Very Much
Variable	*n* (%)	*n* (%)	*n* (%)	*n* (%)	*n* (%)	*n* (%)
**Phase 1 and 2: Wish Capture/Design**						
Joy and Happiness	10 (0.9%)	27 (2.4%)	53 (4.6%)	101 (8.8%)	252 (22%)	704 (61.4%)
Individual Well-being	9 (0.8%)	30 (2.6%)	66 (5.8%)	128 (11.2%)	330 (28.9%)	579 (50.7%)
Family Bonding	11 (1%)	20 (1.8%)	64 (5.6%)	129 (11.3%)	276 (24.2%)	642 (56.2%)
Respite and Distraction	15 (1.3%)	38 (3.3%)	60 (5.2%)	155 (13.5%)	323 (28.2%)	553 (48.3%)
Inclusion and Social Engagement	14 (1.2%)	42 (3.7%)	79 (6.9%)	193 (16.9%)	360 (31.5%)	456 (39.9%)
Looking forward	10 (0.9%)	34 (3%)	52 (4.5%)	143 (12.5%)	314 (27.4%)	594 (51.8%)
Self-efficacy	11 (1%)	25 (2.2%)	66 (5.8%)	158 (13.8%)	308 (26.9%)	577 (50.4%)
**Phase 3: Wish Anticipation**						
Joy and Happiness	10 (0.9%)	4 (0.3%)	33 (2.9%)	84 (7.3%)	275 (24%)	739 (64.5%)
Individual Well-being	9 (0.8%)	10 (0.9%)	44 (3.9%)	124 (10.9%)	329 (28.8%)	626 (54.8%)
Family Bonding	11 (1%)	7 (0.6%)	45 (4%)	119 (10.5%)	315 (27.7%)	641 (56.3%)
Respite and Distraction	13 (1.1%)	15 (1.3%)	40 (3.5%)	144 (12.6%)	382 (33.5%)	546 (47.9%)
Inclusion and Social Engagement	15 (1.3%)	16 (1.4%)	63 (5.5%)	187 (16.4%)	366 (32.2%)	490 (43.1%)
Looking forward	7 (0.6%)	12 (1%)	46 (4%)	132 (11.5%)	347 (30.4%)	599 (52.4%)
Self-efficacy	8 (0.7%)	9 (0.8%)	45 (4%)	147 (12.9%)	359 (31.5%)	570 (50.1%)
**Phase 4: Wish Realization**						
Joy and Happiness	5 (0.4%)	3 (0.3%)	11 (1%)	17 (1.5%)	81 (7.1%)	1029 (89.8%)
Individual Well-being	6 (0.5%)	2 (0.2%)	17 (1.5%)	50 (4.4%)	166 (14.5%)	903 (78.9%)
Family Bonding	7 (0.6%)	5 (0.4%)	18 (1.6%)	51 (4.5%)	163 (14.3%)	899 (78.7%)
Respite and Distraction	8 (0.7%)	12 (1.1%)	22 (1.9%)	72 (6.3%)	199 (17.4%)	829 (72.6%)
Inclusion and Social Engagement	9 (0.8%)	15 (1.3%)	34 (3%)	103 (9%)	258 (22.6%)	722 (63.3%)
Looking forward	8 (0.7%)	6 (0.5%)	17 (1.5%)	68 (6%)	203 (17.8%)	840 (73.6%)
Self-efficacy	11 (1%)	4 (0.4%)	19 (1.7%)	73 (6.4%)	209 (18.4%)	819 (72.2%)
**Phase 5: Short-Term Impact**						
Joy and Happiness	8 (0.7%)	8 (0.7%)	10 (0.9%)	38 (3.3%)	164 (14.4%)	914 (80%)
Individual Well-being	10 (0.9%)	10 (0.9%)	16 (1.4%)	57 (5%)	261 (22.8%)	790 (69.1%)
Family Bonding	10 (0.9%)	8 (0.7%)	21 (1.8%)	52 (4.5%)	233 (20.4%)	820 (71.7%)
Respite and Distraction	10 (0.9%)	17 (1.5%)	21 (1.8%)	96 (8.4%)	245 (21.5%)	753 (65.9%)
Inclusion and Social Engagement	17 (1.5%)	18 (1.6%)	26 (2.3%)	124 (10.9%)	301 (26.4%)	654 (57.4%)
Looking forward	13 (1.1%)	14 (1.2%)	18 (1.6%)	75 (6.6%)	244 (21.4%)	776 (68.1%)
Self-efficacy	11 (1%)	10 (0.9%)	13 (1.1%)	76 (6.7%)	265 (23.2%)	766 (67.1%)
**Phase 5: Long-Term Impact**						
Joy and Happiness	6 (0.5%)	7 (0.6%)	10 (0.9%)	38 (3.3%)	156 (13.6%)	926 (81%)
Individual Well-being	8 (0.7%)	9 (0.8%)	19 (1.7%)	65 (5.7%)	274 (24%)	766 (67.1%)
Family Bonding	7 (0.6%)	11 (1%)	18 (1.6%)	74 (6.5%)	250 (21.9%)	780 (68.4%)
Respite and Distraction	9 (0.8%)	19 (1.7%)	27 (2.4%)	100 (8.8%)	260 (22.9%)	721 (63.5%)
Inclusion and Social Engagement	9 (0.8%)	18 (1.6%)	41 (3.6%)	114 (10.1%)	306 (27%)	646 (57%)
Looking forward	9 (0.8%)	10 (0.9%)	31 (2.7%)	75 (6.6%)	260 (22.8%)	757 (66.3%)
Self-efficacy	11 (1%)	7 (0.6%)	21 (1.8%)	85 (7.4%)	256 (22.4%)	762 (66.7%)

**Table 6 children-12-00047-t006:** Negative outcomes reported by child respondents (*n* = 535).

	*n* (%)
**Felt sadness**	
Do not remember	9 (1.68%)
Not at all	267 (49.91%)
Not really	70 (13.08%)
Neutral	54 (10.09%)
A little	91 (17.01%)
Very much	42 (7.85%)
Missing	2 (0.37%)
**Felt disappointment**	
Do not remember	6 (1.12%)
Not at all	288 (53.83%)
Not really	72 (13.46%)
Neutral	72 (13.46%)
A little	54 (10.09%)
Very much	37 (6.92%)
**Felt frustration**	
Do not remember	7 (1.3%)
Not at all	286 (53.46%)
Not really	64 (11.96%)
Neutral	55 (10.28%)
A little	77 (14.39%)
Very much	43 (8.04%)

**Table 7 children-12-00047-t007:** Secondary outcomes by child and parent respondents.

	Children	Parents/Guardians
	Count (%)	Count (%)
**Created a lasting memory of a happy occasion**		
Do not remember	2 (0.37%)	3 (0.28%)
Strongly disagree	1 (0.19%)	5 (0.47%)
Disagree	3 (0.56%)	10 (0.94%)
Undecided	13 (2.43%)	0 (0.00%)
Agree	197 (36.82%)	268 (25.24%)
Strongly agree	314 (58.69%)	724 (68.17%)
Missing	5 (0.93%)	52 (4.90%)
**Created a sense of normalcy**		
Do not remember	14 (2.62%)	7 (0.66%)
Strongly disagree	4 (0.75%)	10 (0.94%)
Disagree	12 (2.24%)	25 (2.35%)
Undecided	35 (6.54%)	0 (0.00%)
Agree	231 (43.18%)	370 (34.84%)
Strongly agree	234 (43.74%)	542 (51.04%)
Missing	5 (0.93%)	108 (10.17%)
**Benefitted other family members**		
Do not remember	9 (1.68%)	8 (0.75%)
Strongly disagree	6 (1.12%)	17 (1.60%)
Disagree	10 (1.87%)	32 (3.01%)
Undecided	22 (4.11%)	0 (0.00%)
Agree	217 (40.56%)	338 (31.83%)
Strongly agree	266 (49.72%)	548 (51.60%)
Missing	5 (0.93%)	119 (11.21%)
**Gave their family relief from the child’s medical care/treatment**		
Do not remember	12 (2.24%)	5 (0.47%)
Strongly disagree	6 (1.12%)	13 (1.22%)
Disagree	13 (2.43%)	21 (1.98%)
Undecided	36 (6.73%)	0 (0.00%)
Agree	186 (34.77%)	317 (29.85%)
Strongly agree	277 (51.78%)	575 (54.14%)
Missing	5 (0.93%)	131 (12.34%)
**Think a wish experience is important for every child with a critical illness to have**		
Do not remember	2 (0.37%)	0 (0.00%)
Strongly disagree	2 (0.37%)	8 (0.75%)
Disagree	1 (0.19%)	1 (0.09%)
I’m not sure	0 (0.00%)	0 (0.00%)
Agree	122 (22.80%)	143 (13.47%)
Strongly agree	396 (74.02%)	870 (81.92%)
Missing	12 (2.24%)	40 (3.77%)

**Table 8 children-12-00047-t008:** Primary outcomes in wish children and parents.

	Child Reporting on Their Own Wish Experience			Parent Reporting on Their Own Wish Experience				Bonferroni Adjusted *p*-Value
	*n*	Median	IQR	*n*	Median	IQR	*p*-Value	
**Phase 1 and 2: Wish Capture/Design**								
Joy and Happiness	535	5	2	1061	5	1	0.000	0.001
Individual Well-being	533	4	2	1059	5	1	0.000	0.000
Family Bonding	534	4	2	1057	5	1	0.000	0.000
Respite and Distraction	535	4	2	1059	4	1	0.000	0.000
Inclusion and Social Engagement	533	4	2	1059	4	2	0.000	0.000
Looking forward	534	4	2	1062	5	1	0.000	0.000
Self-efficacy	531	4	2	1060	5	1	0.000	0.000
**Phase 3: Wish Anticipation**								
Joy and Happiness	534	5	1	1060	5	1	0.001	0.043
Individual Well-being	532	4	2	1058	5	1	0.000	0.000
Family Bonding	533	4	2	1053	5	1	0.000	0.000
Respite and Distraction	534	4	2	1055	4	1	0.000	0.000
Inclusion and Social Engagement	534	4	2	1052	4	1	0.000	0.000
Looking forward	535	4	2	1058	5	1	0.000	0.000
Self-efficacy	534	4	2	1052	4	1	0.000	0.000
**Phase 4: Wish Realization**								
Joy and Happiness	534	5	0	1060	5	0	0.711	1.000
Individual Well-being	533	5	1	1058	5	0	0.007	0.228
Family Bonding	535	5	0	1057	5	0	0.328	1.000
Respite and Distraction	533	5	1	1056	5	1	0.393	1.000
Inclusion and Social Engagement	533	5	1	1055	5	1	0.857	1.000
Looking forward	533	5	0	1056	5	1	0.106	1.000
Self-efficacy	530	5	1	1049	5	1	0.746	1.000
**Phase 5: Short-Term Impact**								
Joy and Happiness	534	5	0	1056	5	0	0.882	1.000
Individual Well-being	535	5	1	1058	5	1	0.805	1.000
Family Bonding	533	5	1	1058	5	1	0.810	1.000
Respite and Distraction	531	5	1	1056	5	1	0.997	1.000
Inclusion and Social Engagement	533	5	1	1054	5	1	0.050	1.000
Looking forward	533	5	1	1055	5	1	0.053	1.000
Self-efficacy	533	5	1	1055	5	1	0.064	1.000
**Phase 5: Long-Term Impact**								
Joy and Happiness	534	5	0	1058	5	0	0.701	1.000
Individual Well-being	530	5	1	1055	5	1	0.036	1.000
Family Bonding	533	5	1	1054	5	1	0.064	1.000
Respite and Distraction	531	5	1	1050	5	1	0.889	1.000
Inclusion and Social Engagement	533	5	1	1048	5	1	0.926	1.000
Looking forward	534	5	1	1056	5	1	0.484	1.000
Self-efficacy	533	5	1	1056	5	1	0.597	1.000

## Data Availability

The data that support the findings of this study are not publicly available due to confidentiality concerns. Researchers interested in accessing the data should contact the corresponding author who will evaluate requests on a case-by-case basis.

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
