# Peer review of "Wish-Granting Interventions Promote Positive Emotions in Both the Short and Long Term in Children with Critical Illnesses and Their Families"

_children, 2024, doi:10.3390/children12010047_

Round 1

Reviewer 1 Report

Comments and Suggestions for Authors

I appreciate the opportunity to review this article and congratulate the authors on their work. I find the article both interesting and significant as it makes a unique contribution to the field of child health. 

This article has two major constraints from my point of view.

The first constraint concerns the children's characteristics. Children between the ages of 13 and 17 were surveyed about experiences that could be up to 5 years old. This means that when some children experienced the wish, they were school-age children and at the time of the survey they were adolescents. Developmental characteristics differ, leading to different perceptions. This should be discussed in both the Introduction and Discussion sections. Also, critical illness is a broad concept that should be defined in the introduction, and although authors refer to “age and disease-specific responses to interventions” as a limitation, this should be discussed more fully in the manuscript.

The second constraint concerns the references used throughout the manuscript. The authors present only 13 references and most of them were published ≥5 years ago. Although the references seem adequate, they are neither sufficient to support the discussion nor up-to-date. I understand that studies in this area are scarce, however, there are related articles that could explain, for example, the importance for families to provide a respite period and the impact that has on quality of life. I suggest a deeper and more thorough search.

While I compliment the authors on their work, addressing these concerns could significantly improve the clarity and robustness of their article. 

Author Response

Reviewer Comment

Response

I appreciate the opportunity to review this article and congratulate the authors on their work. I find the article both interesting and significant as it makes a unique contribution to the field of child health. 

We would like to thank the reviewer for taking the time to read and give feedback on the manuscript.

This article has two major constraints from my point of view.

The first constraint concerns the children's characteristics. Children between the ages of 13 and 17 were surveyed about experiences that could be up to 5 years old. This means that when some children experienced the wish, they were school-age children and at the time of the survey they were adolescents. Developmental characteristics differ, leading to different perceptions. This should be discussed in both the Introduction and Discussion sections.

Thank you for the suggestion, we agree that this is an important point that should be acknowledged. We have added a short discussion on the potential role of puberty in affecting the perception of a granted wish, and how puberty develops differently between groups. In the future this would be an interesting aspect to take into account when researching this age range.

Also, critical illness is a broad concept that should be defined in the introduction, and although authors refer to “age and disease-specific responses to interventions” as a limitation, this should be discussed more fully in the manuscript.

A definition of a critical illness has been added in the introduction, see lines 35-36.

We agree that we can expand on the differences between wish children and their families that could influence the impact of a wish. This has been expanded on in lines 350-385.

The second constraint concerns the references used throughout the manuscript. The authors present only 13 references and most of them were published ≥5 years ago. Although the references seem adequate, they are neither sufficient to support the discussion nor up-to-date. I understand that studies in this area are scarce, however, there are related articles that could explain, for example, the importance for families to provide a respite period and the impact that has on quality of life. I suggest a deeper and more thorough search.

We have added several new references to our manuscript that examine both wish-granting interventions and psychosocial interventions more generally.

While I compliment the authors on their work, addressing these concerns could significantly improve the clarity and robustness of their article. 

Thank you for your feedback on the article, we believe with your amendments it is much improved.

Reviewer 2 Report

Comments and Suggestions for Authors

Dear Authors,

Congratulations on the work done.

I am sending my suggestions for improvement.

The summary is presented in a way that aligns with the various sections that should be included in a summary, allowing readers to understand the content of the article.

The introduction contextualizes the study, providing a general overview of the topic the authors aim to address. However, it would be relevant for the authors to highlight the gaps that may exist around this topic. I believe that exploring aspects such as the psychological impact of moments of happiness on children’s well-being, among others, could be relevant to enhance the subject under analysis.

In the methods section, aspects such as the lack of monitoring of the total number of invited participants in the study are noted, which may compromise the interpretation of response rates and the representativeness of the data. Moreover, the authors should justify the rationale behind the three questionnaire application methods to determine whether there is any risk of response bias. Additionally, they should explain why children under the age of 13 were excluded.

The results are presented clearly and in detail, with extensive tables showing precise data on the emotions experienced at different stages of the intervention.

The discussion generally addresses the results well. However, I felt the need for a deeper exploration of the interpretation of negative results, providing justifications for their occurrence and comparing them with international studies, while also addressing cultural
and regional diversity issues. The authors should further explore recommendations for practice, offering specific guidance on how to incorporate these interventions into daily practice.

The conclusion is clear and effectively reinforces the main findings, highlighting the importance of interventions for critically ill children and their families.

Author Response

Reviewer Comment

Response

Dear Authors,

Congratulations on the work done.

I am sending my suggestions for improvement.

Objective and Relevance: The article addresses the effects of wish-granting interventions for children with critical illnesses and their families. The research focuses on both the immediate and long-term effects of the experiences provided by these interventions. It is a relevant topic, contributing to a comprehensive perspective on the impact of such interventions in different cultural and geographical contexts.

We would like to thank the reviewer for taking the time to read and give feedback on the manuscript.

Methodology: The research is robust and based on a significant sample, with the data collection procedures and statistical analysis well described. I suggest that the authors provide more detail on how the data were handled to minimize potential biases, especially considering the different levels of internet access and literacy in some countries.

We have provided additional details in the manuscript on the steps taken to minimize potential biases. First, the survey was translated into many different languages, either by a professional translator or an Affiliate staff member, and subsequently reviewed by a second staff member (a native speaker). This ensured that the survey was understandable for our respondents. Further, we explain that a visual aid was provided in terms of an image of the Wish Journey, to improve participant recall of the stages of their experience (see lines 100-104). In addition, it is already stated in the manuscript that the survey was sometimes conducted by phone or in phone by Affiliate staff, in order to expand participation to otherwise underrepresented groups. Nevertheless, we acknowledge residual biases may persist in terms of literacy and internet access (see Discussion section).

We have also added more information on the handling of the data to reduce bias, namely that cases with implausible data related to the age of the child or the time since the wish was granted were removed. This prevented low quality data from affect the results. See lines 154-155.

Results Analysis: The results are presented clearly, with an emphasis on the positive emotions experienced by the children and parents during and after the intervention. The statistical analysis is appropriate, using the Wilcoxon test to compare the participants' responses. The comparison between the experiences of the children and the parents during the different phases of the intervention (wish capture, anticipation, realization, short- and long-term impact) is interesting and well executed.

We would like to thank the reviewer for their feedback on the results section.

Discussion: The discussion appropriately contextualizes the results, aligning them with existing literature. The explanation of the "fading affect bias" (FAB), which suggests that positive emotions last longer than negative ones, is a good point for understanding the longevity of the benefits of wish-granting interventions. Although the study covered 24 countries, there is no detailed analysis of how different cultural and regional contexts may have influenced the participants' responses. I would suggest including a more in-depth reflection on how the experiences of children and parents may vary depending on culture, social norms, and healthcare practices in each region, as this could enrich the discussion.

We agree that we can expand on the differences between wish children and their families that could influence the impact of a wish. We have added to the discussion to include recognition of multiple factors that could interact to influence the impact of a wish, namely age, illness, wish type, family circumstances, culture and geographic region. See lines 350-385.

Strengths and Limitations: The main strength of the study is its international approach and the relatively large sample, which enhances the generalizability of the results. The limitations are well acknowledged, including recall bias, differences in survey implementation across countries, and the lack of data from younger age groups or specific types of illness. These limitations are common in large-scale studies and indeed suggest areas for improvement, such as the need for a prospective study. I believe this section could be expanded by detailing how cultural differences may have influenced the participants' responses.

The role of cultural differences, social norms and healthcare practices in shaping the impact of a wish is now discussed in lines 350-385.

Conclusion: The conclusion reinforces the importance of wish-granting interventions for children with critical illnesses and their families, highlighting the benefits both in the short and long term. The recommendation to incorporate these interventions into clinical routines is a valuable practical contribution to the field of pediatric healthcare and palliative care.

Thank you for your feedback on the article, we believe with our amendments it is much improved.

Reviewer 3 Report

Comments and Suggestions for Authors

This is a well written manuscript, with good flow. A pleasure to read. There are, however, some improvements needed in the analytical section, which are presented below.

Can results of Table 4&5 be presented in graph? Given that frequencies are reported for each and every category, and there are plenty, the changes between stages are incomprehensible – one cannot make analytical statements from the table alone, plus descriptive statistics should be added like means (or median) and standard variations at least.

It seems that only bivariate analysis is conducted in table 8. One would expect at least controlling for the demographic characteristics of families (like wealth) and or of children. Plus, the changes from short-term to long-term changes. Given the international nature of data some sort of cross-national analysis is needed, either observing differences across countries or groupings of countries.

Author Response

Reviewer Comment

Response

This is a well written manuscript, with good flow. A pleasure to read. There are, however, some improvements needed in the analytical section, which are presented below.

We would like to thank the reviewer for taking the time to read and give feedback on the manuscript.

Can results of Table 4&5 be presented in graph? Given that frequencies are reported for each and every category, and there are plenty, the changes between stages are incomprehensible – one cannot make analytical statements from the table alone, plus descriptive statistics should be added like means (or median) and standard variations at least.

Two graphs (Figures 1 and 2) have been produced for the manuscript that show the proportion of children and parents answering ‘a little’ or ‘very much’ to the primary outcome questions, indicating presence of a positive emotion to some extent. We believe this aids the interpretation of the table and the accompanying text.

The median and IQR for the primary outcomes are shown in Table 8 so we have not added these to Tables 4 and 5 to avoid redundancy.

It seems that only bivariate analysis is conducted in table 8. One would expect at least controlling for the demographic characteristics of families (like wealth) and or of children. Plus, the changes from short-term to long-term changes. Given the international nature of data some sort of cross-national analysis is needed, either observing differences across countries or groupings of countries.

We fully agree with the reviewer that examining this data through the lens of wealth or educational background would be very interesting and valuable in advancing our understanding of the impact of a wish. Demographic data beyond age and gender were not collected for children so it is not possible to control for this in this study. However, we certainly see this as an avenue for future research and therefore have added a note about this in the Discussion section (see lines 350-381).

Regarding an analysis of short-term to long-term changes, we refrained from doing this in this study as the definition of ‘long-term’ is fairly varied between participants (between 0 and 5 years). For this reason it would be difficult to draw valid results. In future we plan to incorporate consistent monitoring of participants in a prospective study in the future (see lines 342-349).

We appreciate the suggestion of conducting a cross-national analysis. However, we decided not to conduct a statistical comparison of regions due to the significant variations in response rates among countries within each region. This raised concerns in terms of disproportionate representation of countries per region, which limits generalizability at the regional level and may give misleading results. We acknowledge this as a limitation to this study in the Discussion section. We will however in the future focus our research amongst others on this interesting point.

Round 2

Reviewer 1 Report

Comments and Suggestions for Authors

The authors have considered the suggestions and comments made in the revised manuscript, which I believe is suitable for publication in its present form.

Author Response

Comment: The authors have considered the suggestions and comments made in the revised manuscript, which I believe is suitable for publication in its present form.

Response: We would like to thank the reviewer for their feedback.

Reviewer 3 Report

Comments and Suggestions for Authors

Thank you very much for responding to the comments and making changes to the manuscript. Especially graphs make analysis of descriptive results much easier. I have still one concern. Given that long-term effects are not examined, I would urge authors to delete "long-term" from title as title is now misleading given the fact that long term effects cannot be examined. 

Author Response

Comment: Thank you very much for responding to the comments and making changes to the manuscript. Especially graphs make analysis of descriptive results much easier. I have still one concern. Given that long-term effects are not examined, I would urge authors to delete "long-term" from title as title is now misleading given the fact that long term effects cannot be examined. 

Response: We would like to thank the reviewer for their feedback. We would prefer to maintain the title as it is as it uses terminology that we define and use consistently throughout the manuscript. We recognise the limitations of this terminology in the Discussion section, and we intend to conduct a prospective study to investigate this further in future.